# Online false discovery rate control for anomaly detection in time series

**Quentin Rebjock**[*]
EPFL

**Barış Kurt**
Amazon Research

**Tim Januschowski**
Amazon Research

**Laurent Callot**
Amazon Research

## Abstract

This article proposes novel rules for false discovery rate control (FDRC) geared towards online anomaly detection in time series. Online FDRC rules allow to control the properties of a sequence of statistical tests. In the context of anomaly detection, the null hypothesis is that an observation is normal and the alternative is that it is anomalous. FDRC rules allow users to target a lower bound on precision in unsupervised settings. The methods proposed in this article overcome short-comings of previous FDRC rules in the context of anomaly detection, in particular ensuring that power remains high even when the alternative is exceedingly rare (typical in anomaly detection) and the test statistics are serially dependent (typical in time series). We show the soundness of these rules in both theory and experiments.

## 1 Introduction

Online anomaly detection is critical for many monitoring applications in health care, IT operations, manufacturing, or retail (e.g., [11, 4, 20]). In such settings, observations of one or several metrics of interest (naturally represented as time series) are sent sequentially to a detector. It is tasked with deciding whether a given observation is anomalous or not. Accordingly, anomaly detection in time-series data is a rich field that has been surveyed in several articles [7, 6], Foorthuis [9] proposes a typology of anomalies, and Sadik and Gruenwald [21] discusses open research questions in the field.

Arguably one of the most common approaches in addressing anomaly detection is via employing an unsupervised probabilistic anomaly scorer that assigns a probability to each point. This translates the problem of anomaly detection into one of online multiple hypotheses testing. Specifically, at each time step $t$, we observe a new data point $z_t$ and emit the null hypothesis $H_t$ that $z_t$ is *not* anomalous. This hypothesis is tested with a $p$-value provided by the scorer and may be rejected (detection of an anomaly) if there is enough evidence against it.

As in any statistical testing problem, a decision threshold determines whether the null hypothesis is rejected or not. In the case of a single test, the most common convention is to report a discovery (a point is anomalous) if the associated $p$-value is smaller than some preselected threshold $\alpha$. The effect is said to be *statistically significant at level $\alpha$* and the probability to falsely declare the point anomalous is less than $\alpha$. Importantly, in the case of multiple hypotheses tests, the threshold $\alpha$ does not provide similar guarantees and the fraction of false positives may be arbitrarily close to 1. This particularly applies to anomaly detection because the proportion of alternative hypotheses tends to be very low. Hence, multiple hypotheses testing requires other mechanisms to set decision thresholds. In the case where all $p$-values are known in advance (offline setting), classical methods typically shrink the threshold $\alpha$ conservatively [2, 3, 27].

In the setting of anomaly detection, a natural quantity to control is the false discovery rate (FDR) as introduced by [2], that is, the ratio of falsely labeled anomalies to total anomalies. Methods for

---

[*]Correspondence: quentin.rebjock@epfl.ch.

sequential FDR control (FDRC) were pioneered by [10] who proposed the so-called $\alpha$-investing strategy, which was later built upon and generalized [1, 15, 16, 18, 19, 30, 34, 16]. Online FDRC methods are appealing for anomaly detection problems as controlling the false discovery rate is equivalent to maximizing recall given a lower bound on precision. Consequently this allows to trade-off precision and recall even in the absence of labelled data, which is another commonality of anomaly detection. All online FDRC methods allocate some threshold at every time step, and reject the hypothesis based on it.

The main contribution of this article is the proposal of a novel method to overcome a common limitation of existing online FDRC methods that renders them unsuited for online anomaly detection tasks. When few rejections are made because the alternative hypothesis is extremely rare, as is the case in most anomaly detection problems, rejection thresholds of existing methods tend to zero. This phenomenon is referred to as $\alpha$-death in the literature [18] and prevents any future discoveries causing a loss of power. To circumvent this problem, we propose a method based on memory decay, revisiting [18], to ensure that rejection thresholds are lower-bounded by a non-zero value. This guarantees that we have power even in the case where alternatives (anomalies) are rare. We demonstrate that this method allows us to control the decaying memory FDR [18], a version of the FDR metric adapted to infinite streams that progressively forgets past decisions. Our approach is fully general in the sense that it can enhance all of the most popular algorithms in the generalized $\alpha$-investing class.

Our second contribution is the adaptation of the local dependency framework [34] to memory decay. This allows to use our methods for practical applications where $p$-values are in general not independent. We evaluate experimentally the performances of the proposed algorithms and demonstrate that we can overcome the challenges occurring when alternative hypotheses are exceedingly rare.

This paper is structured as follows. Section 2 formalizes the problem of multiple hypotheses testing for anomaly detection and defines the false discovery rate. Section 3 reviews the main methods for online FDRC and illustrates their limitations. Section 4 details our contributions to prevent $\alpha$-death. Section 5 proposes an adjustment to make the algorithms robust to local dependency in the $p$-values. Finally, Section 6 shows experimentally that our method preserves power while controlling the false discovery rate in anomaly detection settings. We conclude in Section 7.

## 2 Problem Formalization: Multiple hypotheses testing for anomaly detection

We formalize the problem statement of online anomaly detection via an unsupervised approach using probabilistic time series models. There is a rich and growing literature describing such probabilistic models (e.g., [5, 14, 24, 23, 26, 25, 8]) well suited for anomaly detection.

**Deriving $p$-values.** At each time step $t$, hypothesis $H_t$ is tested based on a $p$-value that must be obtained from past observations only. The concept of anomaly is ambiguous and suffers from a subjective and imprecise definition [9]. We assume that anomalies are characterized by statistically rare events that deviate significantly from expectations. For this reason, it is natural to address anomaly detection through *probabilistic* forecasting. A probabilistic forecast is an estimation of the probability distribution of $z_t$ given the past, that is $\mathbf{Prob}\{z_t \mid z_{t'}, t' < t\}$, or equivalently its c.d.f. $F_t$. Under the null hypothesis (no anomaly), such a forecast should be a decent estimate of $z_t$. From the forecast we can derive a $p$-value $p_t$, which is the probability to observe an event in the tail of the distribution relative to $z_t$ under the assumption that $H_t$ is null. Hence, a small value for $p_t$ is an indication that $H_t$ must be rejected. We may derive two-sided $p$-values as $p_t = 2 \min \left( F_t(z_t), 1 - F_t(z_t) \right)$ using the predicted c.d.f. $F_t$ for symmetric distributions.

With this principle, we can infer a sequence of $p$-values $\{p_t\}_{t=1}^{\infty}$ iteratively as new observations are made available. Instead of two-sided $p$-values, other scenarios are conceivable as for certain usages, it may make sense to define anomalies as values either extremely high or low compared to the median. For example, in a scenario where we monitor the resources usage of a compute service, only large values should result in an alert. Accordingly, one-sided $p$-values can be derived to account for only one end of the distribution. More generally, $p$-values need only to be stochastically larger than the uniform distribution under the null hypothesis, that is:

$$\text{if } H_t \text{ is truly null then } \mathbf{Prob}\{p_t \leq u\} \leq u \text{ for all } u \in [0, 1]. \tag{1}$$

With this definition of anomalies, we can decompose the problem of anomaly detection into two distinct sub-problems: *(i)* obtaining $p$-values in line with the application via a forecasting model and *(ii)* using a criterion on the $p$-values to accept/reject hypotheses. We will assume time series models to be given and focus on *(ii)*, that is, the choice of decision thresholds.

**False discovery rate control.** As we reject hypotheses we want to make sure that we do not falsely report too many anomalies. If there was a single hypothesis we could reject it if the associated $p$-value is smaller than some threshold $\alpha$: this ensures that the probability to report an anomaly when it is not is less than $\alpha$. This is an immediate consequence of the definition of the $p$-value in Equation (1). However, this procedure does not provide similar guarantees for a set of hypotheses in general and for sequentially dependent $p$-values as in the case of online anomaly detection in time series in particular. For this reason, in multiple hypotheses settings rejections are based on procedures controlling metrics pertaining to a set of tests.

One of these metrics is the family-wise error rate (FWER) [13], which is the probability to make at least one false discovery. The probability to reject a truly null hypothesis approaches 1 when the number of tests grows. This makes the FWER metric too conservative, leading to very few discoveries for long (possibly infinite) streams of data.

The false discovery rate (FDR) is the more natural quantity to control in our setting. Procedures controlling the FDR aim at maximizing the number of true discoveries subject to an upper bound on the expected proportion of false discoveries. This is equivalent to maximizing recall subject to a user-defined lower bound on precision. For a given set of hypotheses the FDR is formally defined as the expected ratio of false positives to total discoveries:

$$\text{FDR} \triangleq \mathbf{E}\left[\text{FDP}\right] \triangleq \mathbf{E}\left[\frac{|\mathcal{H}^0 \cap \mathcal{R}|}{|\mathcal{R}| \vee 1}\right],$$

where $\mathcal{H}^0$ and $\mathcal{R}$ are the sets of truly null hypotheses and rejections respectively, and FDP is the false discovery proportion of the samples.

This article is concerned with the problem of online FDR control where observations arrive sequentially and we want to make a decision immediately on whether to classify the observation as anomalous or not. Alternative settings have been investigated in the literature. Rather than making a decision at every time step, Zrnic et al. [33] considers testing sequences of hypotheses by batches of variable length. This takes advantage of a partial ordering of the $p$-values at the cost of a delay in the decision, which is not desirable in online anomaly detection. Another option is to aggregate a series of $p$-values [17, 12] and perform FDR control consequently, either sequentially or based on multivariate [22] forecasts. Wang [31] propose to perform online anomaly detection by coupling a STL decomposition with a FDR method based on the distribution of $z$ statistics or the residuals building on a method proposed by Sun and Cai [29].

## 3 Background: Online FDR control

Online testing algorithms aim at controlling a time-dependent variant of the FDR that we define as follows. We let $R_t$ denote the rejection indicator at time $t$ and define the number of rejections $R(T)$ and of the number of false positives $V(T)$ up to time $T$ as

$$R(T) = \sum_{t=1}^{T} R_t \quad \text{and} \quad V(T) = \sum_{t=1}^{T} R_t \mathbf{1}\{t \in \mathcal{H}^0\}.$$

Naturally, we define the FDP and FDR at time $T$ as $\text{FDP}(T) = \frac{V(T)}{R(T)\vee 1}$ and $\text{FDR}(T) = \mathbf{E}\left[\text{FDP}(T)\right]$ respectively.

Online decision rules that control the FDR allocate a decision threshold $\alpha_t$ to test hypothesis $H_t$ at every time step $t$. The hypothesis is rejected if the associated $p$-value is less than $\alpha_t$, and the rejection indicator is $R_t = \mathbf{1}\{p_t \leq \alpha_t\}$. We wish to pick the rejection thresholds $\{\alpha_t\}_{t=1}^{\infty}$ in such a way that $\text{FDR}(T) \leq \alpha$ at any time $T$ for some FDR target $\alpha \in [0,1]$. Decision thresholds must depend solely on past observations, so that for any time $t$ there is a function $f_t$ such that

$$\alpha_t = f_t(R_1, \ldots, R_{t-1}) \in \mathcal{F}^{t-1}, \tag{2}$$

where $\mathcal{F}^t \triangleq \sigma(R_1, \dots, R_t)$ is the $\sigma$-field (information known) at time $t$.

In this section, we make the standard conditional super-uniformity assumption that

$$\text{if } H_t \text{ is truly null then } \mathbf{Prob}\{p_t \leq u \mid \mathcal{F}^{t-1}\} \leq u \text{ for all } u \in [0, 1], \tag{3}$$

which is satisfied if $\{p_t\}_{t=1}^\infty$ is a sequence of valid $p$-values and if null $p$-values are independent of all others. The independence assumption can be questionable for time series data in certain settings. We propose a method to relax it in Section 5 but we maintain it until then to simplify the exposition.

**Generalized $\alpha$-investing.** The first algorithm controlling (a modified version of) the FDR was introduced by [10] with the idea of $\alpha$-investing. This algorithm is part of a broader class of algorithms known as generalized $\alpha$-investing (GAI) rules [1]. Algorithms of the GAI class record a quantity $w_t$ called the wealth, that represents the allocation potential for future decision thresholds. They start with an initial wealth $w_0 \in (0, \alpha)$ and at every time step $t$, an amount $\phi_t$ is spent from the remaining wealth in order to test the hypothesis $H_t$ at level $\alpha_t$. When a rejection occurs, an amount of $\psi_t$ is earned back, which induces the update $w_t = w_{t-1} - \phi_t + R_t \psi_t$. The wealth must always be non-negative, which imposes $\phi_t \leq w_{t-1}$, and the (modified) FDR is controlled only if the quantities $\psi_t$ are bounded adequately. The exact FDR can be controlled by algorithms called LORD and LOND [15]. These algorithms are part of the GAI framework, and satisfy a *monotonic* property, meaning that $f_t$ is a coordinate-wise non-decreasing function in Equation (2). In fact, any monotonic GAI rule controls the FDR, under the assumption that $p$-values are independent [16].

**Oracle rules.** Monotonic GAI rules can be seen as a special case of decision rules relying oracle estimates of the FDP [18, 19, 30]. Maintaining the quantity

$$\text{FDP}^\star(T) = \frac{\sum_{t \leq T, t \in \mathcal{H}^0} \alpha_t}{R(T) \vee 1}$$

below $\alpha$ at time $T$ guarantees that $\text{FDR}(T) \leq \alpha$. This oracle cannot be evaluated as the set of null hypotheses $\mathcal{H}^0$ is not known. A simple way to upper-bound $\text{FDP}^\star$ is to remove the condition $t \in \mathcal{H}^0$ in the summation, which leads to a class of algorithms known as LORD [15, 16, 18]. More refined upper-bounds give rise to adaptive algorithms known as SAFFRON and ADDIS [19, 30]. In short, LORD can be seen as an online version of the Benjamini-Hochberg procedure [2], SAFFRON as an online version of [27], and ADDIS is able to slightly improve on SAFFRON's performances when null $p$-values are not exactly uniformly distributed, but stochastically larger than uniform. The algorithms SAFFRON and ADDIS are able to improve slightly on LORD's power when the proportion of alternative hypotheses is large, typically more than $50\%$, which is not the case in anomaly detection. We illustrate this in Figure 1. For this reason we focus on LORD and provide a more detailed discussion about SAFFRON and ADDIS in Appendix B.

The oracle estimate of LORD is given by

$$\widehat{\text{FDP}}_{\text{LORD}}(T) = \frac{\sum_{t \leq T} \alpha_t}{R(T) \vee 1}$$

and offers a lot of flexibility. FDR control at time $T$ holds for any sequence $\{\alpha_t\}_{t=1}^\infty$ chosen such that $\widehat{\text{FDP}}(T) \leq \alpha$, provided that the functions $f_t$ (in Equation 2) are coordinate-wise non-decreasing. We show here how to generate such a sequence as suggested in [18]. We let $\{\gamma_t\}_{t=1}^\infty$ denote a non-increasing sequence summing to 1, where we set $\gamma_t = 0$ for all $t \leq 0$ for convenience. We also let $\rho_j = \min\{t \geq 0 \mid \sum_{i=1}^t R_i \geq j\}$ denote the time of the $j$th rejection (and $\infty$ if it doesn't exist). Then the sequence defined by

$$\alpha_t = w_0(\gamma_t - \gamma_{t-\rho_1}) + \alpha \sum_{j \geq 1} \gamma_{t-\rho_j} \tag{4}$$

satisfies the FDR control requirements. Note that thresholds decrease to zero except when new rejections are made.

Using the same notations, a special instance of ADDIS is given by the sequence

$$\alpha_t = (\tau - \lambda)\left(w_0(\gamma_{S_0(t)} - \gamma_{S_1(t)}) + \sum_j \gamma_{S_j(t)}\right) \wedge \lambda, \tag{5}$$

for some fixed parameters $1 > \tau > \lambda > 0$, and where we define

$$S_j(t) \triangleq \mathbf{1}\{t > \rho_j\} + \sum_{i=\rho_j+1}^{t-1} \mathbf{1}\{\lambda < p_i \leq \tau\}\,.$$

The corresponding SAFFRON algorithm is recovered by setting $\tau = 1$.

**Limitations of FDR rules.** Anomalies are by definition rare events and the proportion of alternative hypotheses in the stream is expected to be very low, typically less than $1\%$ in practice. This induces two distinct problems for the discovery of anomalies:

P.1 The lower the proportion of alternatives, the harder it is to distinguish them from nulls with extreme values. This difficulty is inherent to FDR control, be it offline or online, and makes anomaly detection challenging.

P.2 In the online setting, the decision thresholds $\alpha_t$ decreases monotonously while no rejection is made. Hence, if no rejection is made for a long time, the thresholds will be essentially null and no additional discovery will occur. This phenomenon is known as $\alpha$-*death* and causes a loss of power in anomaly detection regimes.

The power of these algorithms depends on $\{\gamma_t\}_{t=1}^{\infty}$ and on the nature of the data since they are general and make minimal assumptions. In the remaining, we use the default and robust sequences derived by [18, 19, 30], that is, $\gamma_t \propto \frac{\log(t \wedge 2)}{t \exp \sqrt{\log(t)}}$ for LORD. In experiments involving SAFFRON and ADDIS we also use standard parameters; see Appendix B for more details. Figure 1 illustrates

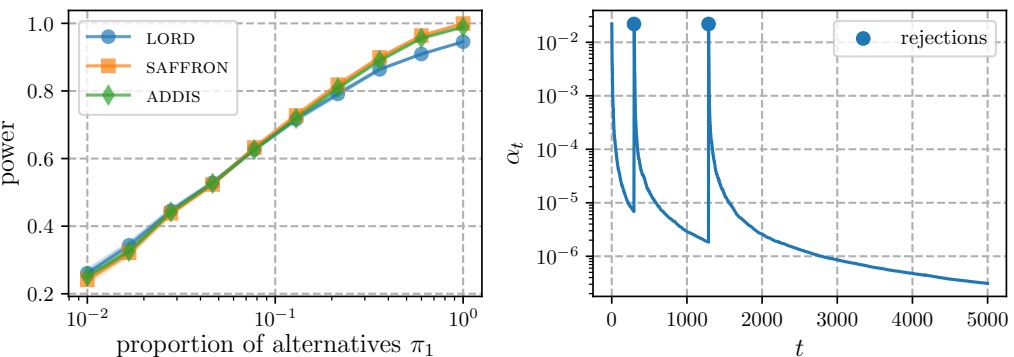

Figure 1: Illustration of the difficulties encountered in FDR control for anomaly detection. *(Left)* The power decreases to zero when the proportion of alternative hypotheses $\pi_1$ is low for all FDR controllers. Stream of artificially generated $p$-values from observations either from $\mathcal{N}(0,1)$ (null) or $\mathcal{N}(3,1)$ (anomaly) with proportion $\pi_1$. *(Right)* SAFFRON's decision thresholds $\alpha_t$ as a function of time for a target $\alpha = 0.1$ and the canonical sequence $\{\gamma_t\}$.

both of these problems. In the regime where the proportion of alternatives is low, LORD, SAFFRON and ADDIS all exhibit the same (poor) performances. Selecting a sequence $\{\alpha_t\}_{t=1}^{\infty}$ going to zero as slowly as possible helps to overcome P.2 to some extent but worsens P.1 in contrast.

# 4 Circumventing $\alpha$-death

In this section, we tackle the issue of $\alpha$-death described in Section 3, which is the biggest practical problem of FDR control for anomaly detection. This suggests a change in the FDR metric, as indicated by [18]. They introduced a decaying memory version of the FDR, meaning that the past is forgotten with a discount factor $\delta \in (0, 1]$.

**Memory decay FDR.** We define decaying memory rejections count $R_\delta(T)$ and false positives count $V_\delta(T)$ as

$$R_\delta(T) = \sum_{t=1}^{T} \delta^{T-t} R_t \quad \text{and} \quad V_\delta(T) = \sum_{t=1}^{T} \delta^{T-t} R_t \mathbf{1}\{t \in \mathcal{H}^0\}.$$

In words, more weight is given to recent rejections, and the past shrinks exponentially. This is arguably the most intuitive notion of FDR for very long, possibly infinite streams of data. Consistently, the decaying memory FDR at level $\delta$ is defined as

$$\text{FDR}_\delta(T) = \mathbf{E}\left[\text{FDP}_\delta(T)\right] = \mathbf{E}\left[\frac{V_\delta(T)}{R_\delta(T) \vee 1}\right]. \tag{6}$$

The LORD rejection thresholds (4) can be adapted as

$$\alpha_t = w_0\left(\delta^{t-\min(\rho_1,t)}\gamma_t - \delta^{t-\rho_1}\gamma_{t-\rho_1}\right) + \alpha \sum_{j \geq 1} \delta^{t-\rho_j}\gamma_{t-\rho_j} \tag{7}$$

in order to control the memory FDR at the desired level $\alpha$ as proposed by [18]. They also show that it inhibits a phenomenon known as *piggybacking* [18], where threshold increments accumulated by previous rejections lead to a sequence of bad decisions.

However, after the first rejection, decision thresholds in Equation (7) suffer from an exponential decay and quickly tend to zero if no rejection is done in the next few steps. The authors suggest that when thresholds are too low, the rejection process should be paused and abstain from making any decision. After some time the oracle estimate decreases because of the exponential decay and the rejection process can be resumed. This scheme poses two problems: *(i)* during an abstention phase we are not able to reject any hypothesis, even those with very low $p$-values that might be critical and *(ii)* there is no theoretical guarantee that the FDR is still controlled after resetting the process with initial values.

In this section we establish one of the key contributions of this paper, showing that the idea of memory decay, allows to avoid $\alpha$-death while controlling the FDR. When no rejection is made both $V_\delta(T)$ and $R_\delta(T)$ decay to zero. Intuitively, this translates into a minimal rejection threshold that doesn't depend on the last rejection time. The standard expression of the FDR contains an asymmetry before and after the first rejection time $\rho_1$ because of the $\vee$ operator in the denominator. For this reason, our results are more naturally expressed in terms of *smoothed* FDR (inspired from [16]), that is

$$\text{sFDR}_\delta(T) = \mathbf{E}\left[\text{sFDP}_\delta(T)\right] = \mathbf{E}\left[\frac{V_\delta(T)}{R_\delta(T) + \eta}\right],$$

for some small smoothing parameter $\eta > 0$. We show in Appendix D how to deal with the case where the quantity to control is the memory FDR at level $\delta$ as defined in Equation (6).

**LORD memory decay.** As in Section 3, we express algorithms in terms of oracle rules. Remember that decision thresholds $\alpha_t = f_t(R_1, \ldots, R_{t-1})$ are functions of the past. We define the memory decay version of the LORD oracle as

$$\widehat{\text{FDP}}_{\text{LORD}}^\delta(T) = \frac{\sum_{t \leq T} \delta^{T-t}\alpha_t}{R_\delta(T) + \eta}.$$

We derive a memory decay analogue of the results in [18].

**Proposition 1.** *Suppose that functions $f_t$ are coordinatewise non-decreasing. If $p$-values satisfy relation 3 then picking decision thresholds $\alpha_t$ such that $\widehat{\text{FDP}}_{\text{LORD}}^\delta(T) \leq \alpha$ at time $T$ ensures that $sFDR_\delta(T) \leq \alpha$.*

Proof given in Appendix C.1.

As for all oracle rules presented above, Proposition 1 gives a lot of freedom regarding the choice of the sequence of decision thresholds. We exhibit here a special instance that naturally generalizes the standard LORD algorithm and that doesn't suffer from $\alpha$-death when $\delta < 1$. Let $\{\tilde{\gamma}_t\}_{t=1}^\infty$ be a positive non-increasing sequence such that $\sum_{t=1}^{T} \delta^{T-t}\tilde{\gamma}_t \leq 1$ for all $T$. Notice that such a sequence

admits a limit no larger than $1 - \delta$, and may be chosen lower-bounded by $1 - \delta$. A natural choice is $\tilde{\gamma}_t = \max(\gamma_t, 1 - \delta)$. Rejection thresholds defined as

$$\alpha_t = \alpha \eta \tilde{\gamma}_t + \alpha \sum_j \delta^{t-\rho_j} \gamma_{t-\rho_j} \tag{8}$$

satisfy the assumptions of Proposition 1. In the case where $\tilde{\gamma}_t = \max(\gamma_t, 1 - \delta)$, this means that there is a minimal rejection threshold, namely $\alpha \eta (1 - \delta)$, no matter how long ago the last rejection occurred.

**SAFFRON and ADDIS memory decay.** Algorithms SAFFRON [19] and ADDIS [30] can be adapted in the same way as LORD in order to avoid $\alpha$-death. We defer precise statements and analysis to Appendix B. We provide here a special instance for some fixed parameters $1 > \tau > \lambda > 0$. The sequence of thresholds defined by

$$\alpha_t = \alpha (\tau - \lambda) \left( \eta \tilde{\gamma}_{S_0(t)} + \sum_j \delta^{t-\rho_j} \gamma_{S_j(t)} \right) \wedge \lambda \tag{9}$$

controls the smoothed memory decaying FDR at level $\alpha$. Here again, in the case where $\tilde{\gamma}_t = \max(\gamma_t, 1 - \delta)$, this means that there is a minimal rejection threshold, namely $\alpha \eta (\tau - \lambda)(1 - \delta) \wedge \lambda$. In Figure 2 we exhibit a qualitative example showing how the minimal rejection threshold helps to

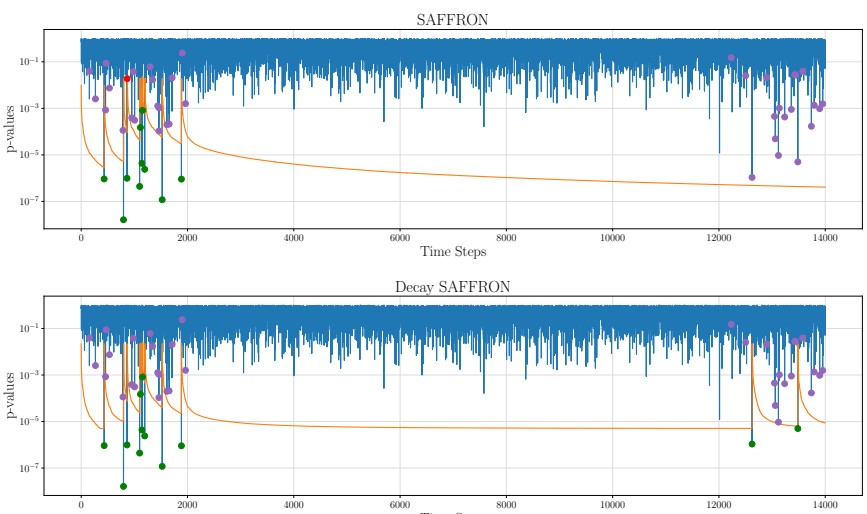

Figure 2: SAFFRON rejection threshold with memory decay (bottom) and without (top). True positives in green, false positives in red, false negatives in purple. Without memory decay, the rejection threshold decreases so much that further discoveries are impossible.

make discoveries after a long time without anomalies.

## 5   Local dependency

In this section, we show how to theoretically overcome the limitation posed by the independence of $p$-values implied by the super-uniformity assumption (3). Time series data contains dependencies that may leak to the $p$-values. If null $p$-values depend on each other the super-uniformity assumption may be broken. We propose a method to deal with the case where null $p$-values are allowed to be locally dependent.

Thresholds of the LOND and LORD algorithms can be adapted to control FDR under arbitrary dependence of $p$-values [15, 16], in analogy to the Benjamini-Yekutieli procedure [3] in the offline setting. It consists essentially in dividing thresholds $\alpha_T$ by the quantity $q(T) \triangleq \sum_{t=1}^T \frac{1}{t}$ at all steps. By extension, SAFFRON and ADDIS could be corrected in a similar way. However, this calibration makes thresholds essentially null in the long term as $q(T)$ goes to infinity. The reason is that arbitrary

dependency of $p$-values is an overly restrictive supposition. As shown in Section 2, the most natural way to derive $p$-values is to generate probabilistic forecast. At time step $t$ such a forecast is typically based on a *context*, that is, a sequence of past observations of finite length. If the dependence in the data is controlled by the forecasting model then the $p$-value $p_t$ is independent on remote $p$-values. More specifically, $p_t$ depends on the last $L_t$ $p$-values only, or, in other words,

$$p_t \perp\!\!\!\perp p_1, \ldots, p_{t-L_t-1}, \tag{10}$$

provided that $L_t$ is large enough. Most commonly, the dependency range is constant over time, that is, $L_t = L$ for all $t$, which we are going to assume. Notice that $L = 0$ and $L = \infty$ recover the independence and arbitrary dependence settings respectively.

The asynchronous hypotheses testing framework established by [34] proposes an approach to control a modified definition of FDR under *local dependency* of $p$-values. We show how to adapt the memory decay LORD algorithm for local dependency in order to control

$$\mathrm{mFDR}(T) = \frac{\mathbf{E}[V_\delta(T)]}{\mathbf{E}[R_\delta(T)] + \eta}.$$

We define the memory decay LORD oracle for locally dependent $p$-values as

$$\widehat{\mathrm{FDP}}_{\mathrm{dep}}^{\delta}(T) = \frac{\sum_{t \leq T} \delta^{T-t} \alpha_t}{R_\delta(T) + \eta}.$$

**Proposition 2.** *Suppose that decision thresholds are monotone with respect to rejection indicators. If $p$-values are locally dependent, that is, satisfy relation (10) then picking decision thresholds $\alpha_t$ such that $\widehat{FDP}_{\mathrm{dep}}^{\delta}(T) \leq \alpha$ ensures that mFDR(T) $\leq \alpha$ at any time $T$.*

Proof given in Appendix C.2. As a special instance of this algorithm we can define

$$\alpha_t = \alpha\eta\tilde{\gamma}_t + \alpha \sum_j \delta^{t-\rho_j-L} \gamma_{t-\rho_j-L}.$$

Notice that increases of the thresholds coming from rejections are delayed by the context length $L$. This procedure controls only a modified version of the FDR but it is known that mFDR behaves like FDR when the number of time steps is large [10].

# 6 Experiments

## 6.1 Simulations

In this section we aim at documenting that our proposed methods are indeed able to control the decay FDR when the proportion of alternative hypothesis (anomalies) decreases while maintaining power. This article does not aim at asserting whether FDRC method can yield higher accuracy than alternative threshold selection methods, such as fixed threshold, for anomaly detection tasks. The main advantage of FDRC method is in being able to target an upper bound on the false discovery rate, and therefore a lower bound on precision, without observing labels during training.

For these experiments we generate data following a mixture of distribution, with a share $\pi_1 \in (0, 1)$ of observations drawn from the anomalous distribution and a share $1 - \pi_1$ drawn from the non-anomalous distribution. We associate a label with each observation, 1 if the observation is drawn from the anomalous distribution, 0 otherwise. We then compute $p$-values of the data under the non-anomalous distribution, mimicking a forecasting-based anomaly scorer while abstracting away model noise and uncertainty.

We apply the different FDRC methods described above to these sequences of $p$-values in order to classify observations as anomalous or not. We use the labels to compute the false discovery rate, decaying false discovery rate, as well as statistical power. We focus here on assessing the pure decay version. The appendix contains further experiments taking local dependency into account.

Figure 3 shows that our methods, labelled `DecayLord` and `DecaySaffron`, are able to control the $\mathrm{FDR}_\delta$ as anomalies become rarer (bottom panel) while maintaining power (top panel) ensuring that a substantial fraction of anomalies are detected. Expectedly, having power against vanishingly rare

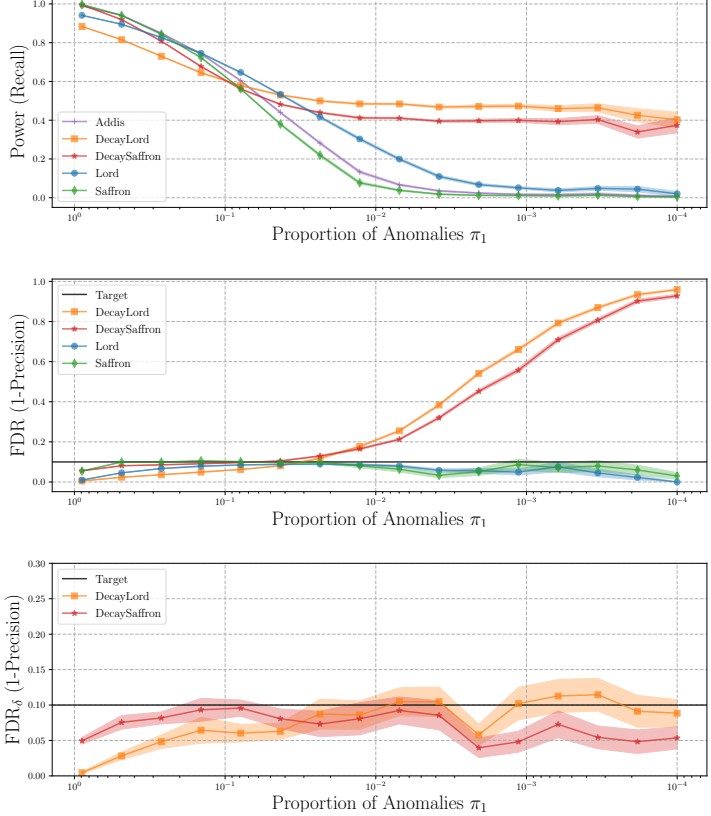

Figure 3: Power (recall) and FDR $(1 - \text{precision})$ changes according to anomaly proportions. For each anomaly proportion, 100 datasets of length 20K are generated by drawing from either $\mathcal{N}(0,1)$ with probability $(1 - \pi_1)$ or from $\mathcal{N}(0,3)$ with probability $\pi_1$ where $\pi_1 \in [10^{-4}, 0.9]$. For all models, the target FDR is 0.1; for SAFFRON, $\lambda = 1/2$, and for ADDIS $\lambda = 1/4$. We set $\delta$ to 0.99 for `DecayLord` and `DecaySaffron` models.

alternatives comes at the cost of a decrease in precision (middle panel). The power (or recall) of non-decay methods tends to zero rapidly as the $\alpha$-death phenomenon prevents them from making any rejections. Their precision, $(1 - \text{FDR})$, remains artificially high since no discovery is ever made.

We provide an additional experiment with artificial data in Appendix A. It compares the Power/FDR curves of our algorithms to popular existing techniques.

## 6.2 Real World Experiments

We next demonstrate the performance of our methods on $p$-values generated by a simple forecaster applied on the Server Machine Dataset (SMD) [28]. The $p$-value for each observation is calculated by assuming that the data follows a Gaussian distribution whose moments are estimated as the sample mean and standard deviation of the previous $n$ points. This naive $p$-value generation method does not attempt to control for dependence in the time-series, potentially introducing dependence in the $p$-values. The SMD is composed of multi-dimensional time series, labels are affixed to the whole dataset for a specific time-stamp rather than to individual series. Therefore, the $p$-values we assign to a given time-stamp is the smallest of the p-values computed on each dimension for that time-stamp. Anomalies are relatively frequent in the SMD, $4.16\%$ of time-stamps are labelled as anomalous. In this setting, both standard and decay version of FDRC rules are expected to perform well.

Figure 4 shows that while both decay and non-decay versions of LORD and SAFFRON do not control the target FDR very precisely (top-left), they control the decay FDR much more accurately (top-

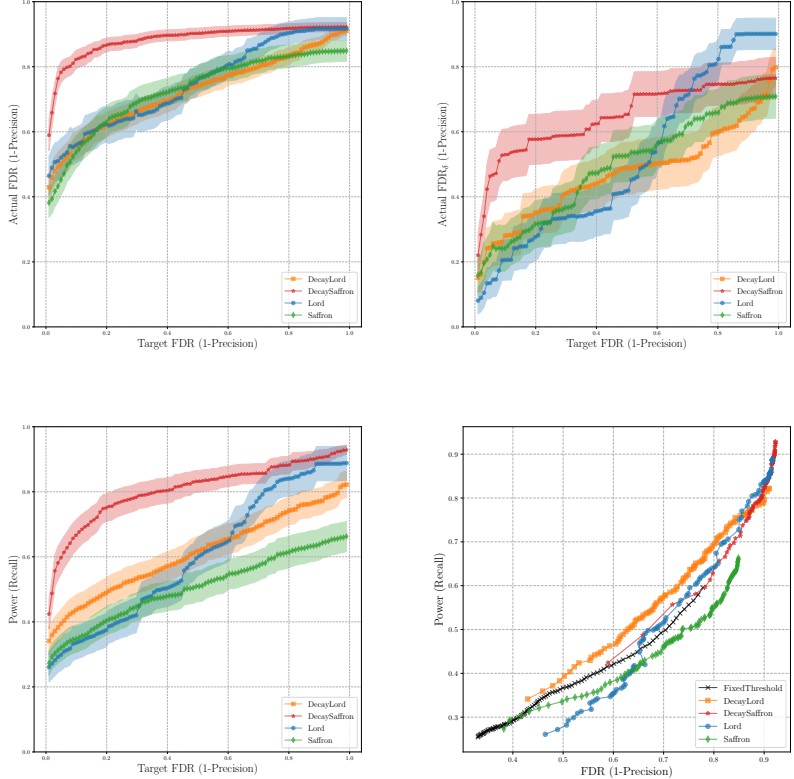

Figure 4: FDR, FDR$_\delta$ and Power(recall) changes according to target FDR over the $p$-values generated by applying a simple forecast algorithm to the Server Machine Dataset. The bottom right plot shows the power change according to actual FDR is given.

right). `DecayLord` and `DecaySaffron` has higher power that non-decay versions when the target FDR is low (bottom-left). For a given, realized false discovery rate, the power of both `DecayLord` and `DecaySaffron` is higher (bottom-right).

## 7 Conclusion, Limitations, and Future Work

This article demonstrates that by using a memory decay procedure, we can ensure that state-of-the-art online false discovery rate control rules can maintain power against rare alternatives. This enables the use of FDRC methods for anomaly detection tasks in streams of time-series data.

This work is limited in its scope to demonstrating in theory and empirically that our modified FDRC rules are effective. We do not consider the problem of tuning the hyper-parameters of these rules, in particular the decay rate $\delta$ as well as the selection of the sequence $\gamma_t$. The choice of these parameters is likely to noticeably influence the performance of the the rules.

We have chosen not to explore the coupling of these rules with anomaly scoring models to evaluate the efficiency of these pairs on real-world anomaly detection tasks as the purpose of this article is on establishing the properties of these rules. In addition, we note that commonly used public data sets to evaluate anomaly detection methods have been convincingly argued [32] to be flawed and not representative of real-world tasks in particular due to an unrealistically high density of anomalies. In such settings, and contrary to many practical problems in anomaly detection, using memory decay FDRC rules would not be useful. We hope to be able to follow up on this work once higher-quality public datasets become available.

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
