# A  Supplementary experiments.

## A.1  Comparison to fixed thresholds

A standard practice in anomaly detection to classify observations is to use a fixed threshold, either over the data itself or over a model-assigned score. In probabilistic anomaly detection, this corresponds to deciding on a $p$-value below which the null hypothesis is rejected. As we argued above, such procedure controls the probability of a false discovery for each individual test but does not provide any control over the whole sequence.

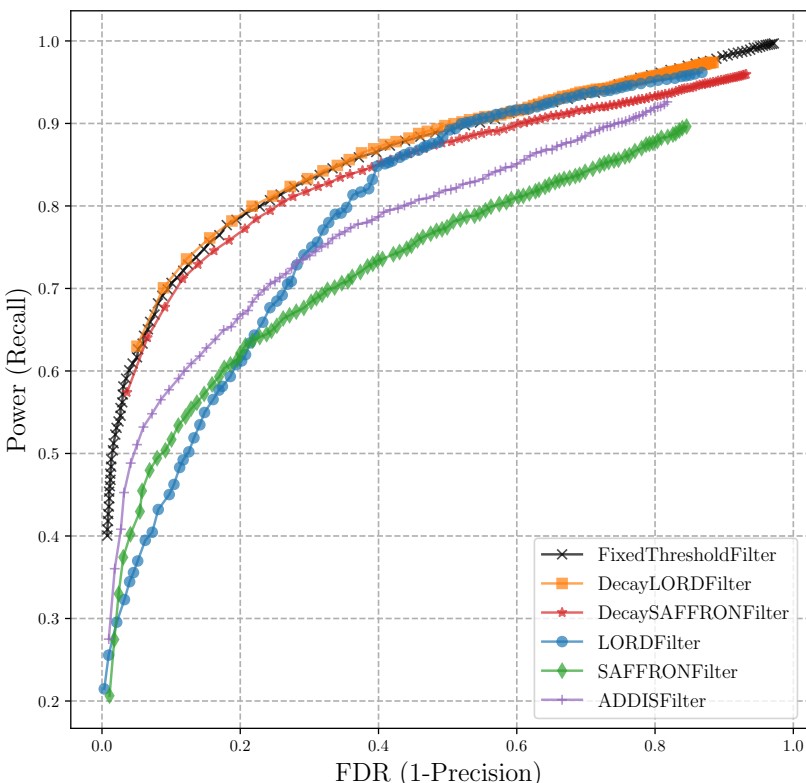

Figure 5: Power (recall) and FDR ($1 - \text{precision}$) for false discovery control rules as well as fixed threshold. The data generating process is the same as in Figure 2.

Figure 5 shows that our decay FDRC rules do draw a similar Precision-Recall curve as the fixed threshold approach does. The major difference between both approaches is that FDRC rules allow users to specify the precision target ex-ante while this is not possible with fixed threshold. Non-decay rules achieve lower recall for a given precision level than both decay FDRC and fixed thresholds on at least part of the precision range.

# B  Discussion about SAFFRON and ADDIS.

## B.1  Standard algorithms

In this section we discuss briefly some of the details in the definitions of SAFFRON and ADDIS [19, 30]. The algorithm LORD cannot be seen as a special case of SAFFRON. However SAFFRON is a

special case of ADDIS as will be presented below. Given two sequences $\{\lambda_t\}_{t=1}^\infty$ and $\{\tau_t\}_{t=1}^\infty$ we define the indicators

$$C_t = \mathbf{1}\{p_t \le \lambda_t\} \qquad \text{and} \qquad K_t = \mathbf{1}\{p_t \le \tau_t\}.$$

We define the filtration

$$\mathcal{F}^t \triangleq \sigma(R_1, \dots, R_t, C_1, \dots, C_t, K_1, \dots, K_t)$$

and require sequences $\{\alpha_t\}_{t=1}^\infty, \{\lambda_t\}_{t=1}^\infty, \{\tau_t\}_{t=1}^\infty$ to be predictable, that is, $\alpha_t, \lambda_t, \tau_t \in \mathcal{F}^{t-1}$. They must also satisfy the inequality $\tau_t > \lambda_t \ge \alpha_t$ for all $t$.

The ADDIS oracle is given by

$$\widehat{\mathrm{FDP}}_{\text{ADDIS}}(T) = \frac{\sum_{t \le T} \alpha_t \frac{\mathbf{1}\{\lambda_t < p_t \le \tau_t\}}{\tau_t - \lambda_t}}{R(T) \vee 1},$$

and SAFFRON can be seen as a special case of ADDIS where $\tau_t = 1$ for all $t$, that is,

$$\widehat{\mathrm{FDP}}_{\text{SAFFRON}}(T) = \frac{\sum_{t \le T} \alpha_t \frac{\mathbf{1}\{\lambda_t < p_t\}}{1 - \lambda_t}}{R(T) \vee 1}.$$

The FDR is controlled at time $T$ if $\widehat{\mathrm{FDP}}_{\text{ADDIS}}(T) \le \alpha$.

A special instance of this algorithm can be derived as follows. We let

$$S_j(t) \triangleq \mathbf{1}\{t > \rho_j\} + \sum_{i=\rho_j+1}^{t-1} \mathbf{1}\{\lambda < p_i \le \tau\},$$

where $\rho_0 = -\infty$ and define

$$\alpha_t = (\tau - \lambda)\Big(w_0(\gamma_{S_0(t)} - \gamma_{S_1(t)}) + \alpha \sum_j \gamma_{S_j(t)}\Big) \wedge \lambda$$

for some $w_0 \in (0, \alpha)$.

In practice, the sequences $\{\lambda_t\}_{t=1}^\infty$ and $\{\tau_t\}_{t=1}^\infty$ are often chosen to be constant. Typical values are $\lambda_t = 1/2$ for SAFFRON and $\lambda_t = 1/4, \tau_t = 1/2$ for ADDIS; they are the ones we choose in our experiments. The sequence $\{\gamma_t\}_{t=1}^\infty \gamma_t \propto t^{-s}$ for both SAFFRON and ADDIS, where $s = 1.6$ unless otherwise specified.

## B.2 Memory decay algorithms

For the memory decay versions controlling $\mathrm{sFDR}_\delta$ we define the oracle

$$\widehat{\mathrm{FDP}}_{\text{ADDIS}}^\delta(T) = \frac{\sum_{t \le T} \delta^{T-t} \alpha_t \frac{\mathbf{1}\{\lambda_t < p_t \le \tau_t\}}{\tau_t - \lambda_t}}{R_\delta(T) + \eta},$$

and have the following result.

**Proposition 3.** *Suppose that quantities $\alpha_t, \lambda_t$ and $1 - \tau_t$ are coordinatewise non-decreasing functions of the past satisfying $\tau_t > \lambda_t \ge \alpha_t$ for all t. If p-values satisfy relation [3] then picking decision thresholds $\alpha_t$ such that $\widehat{\mathrm{FDP}}_{\text{ADDIS}}^\delta(T) \le \alpha$ at time T ensures that $\mathrm{sFDR}_\delta(T) \le \alpha$.*

Proof is given in Appendix C.3. For some fixed parameters $\tau > \lambda$ we can define the following special instance:

$$\alpha_t = \alpha(\tau - \lambda)\Big(\eta\tilde{\gamma}_{S_0(t)} + \sum_j \delta^{t-\rho_j}\gamma_{S_j(t)}\Big) \wedge \lambda.$$

In Appendix D we show how to adapt the statements in order to control the standard memory decay FDR.

## C Omitted proofs

Our proofs are based on the two following lemmas, proposed by [18] and [30].

**Lemma C.1** ([18]). *Suppose that the sequence $\{p_t\}_{t=1}^\infty$ is composed of independent p-values. Let $f_t$ be a sequence of non-decreasing functions such that $\alpha_t = f_t(R_1, \ldots, R_{t-1})$. Then for any non-decreasing function $h$,*

$$\mathbf{E}\left[\frac{\alpha_t}{h(R_{1:T})} \mid \mathcal{F}^{t-1}\right] \geq \mathbf{E}\left[\frac{\mathbf{1}\{p_t \leq \alpha_t\}}{h(R_{1:T})} \mid \mathcal{F}^{t-1}\right],$$

*where $\mathcal{F} = \sigma(R_1, \ldots, R_t)$.*

**Lemma C.2** ([30]). *Suppose that the sequence $\{p_t\}_{t=1}^\infty$ is composed of independent p-values. let $f_t, g_t, \tilde{g}_t$ be three sequences of non-decreasing functions such that $\alpha_t = f_t(R_{1:t-1}, C_{1:t-1}, K_{1:t-1})$, $\lambda_t = g_t(R_{1:t-1}, C_{1:t-1}, K_{1:t-1})$, $\tau_t = \tilde{g}_t(R_{1:t-1}, C_{1:t-1}, K_{1:t-1})$ and $\alpha_t \leq \lambda_t < \tau_t$. Then for any non-decreasing function $h$,*

$$\mathbf{E}\left[\frac{\alpha_t \mathbf{1}\{\lambda_t < p_t \leq \tau_t\}}{(\tau_t - \lambda_t)h(R_{1:T})} \mid \mathcal{F}^{t-1}, K_t = 1\right] \geq \mathbf{E}\left[\frac{\alpha_t}{\tau_t h(R_{1:T})} \mid \mathcal{F}^{t-1}, K_t = 1\right]$$

$$\geq \mathbf{E}\left[\frac{\mathbf{1}\{p_t \leq \alpha_t\}}{h(R_{1:T})} \mid \mathcal{F}^{t-1}, K_t = 1\right],$$

*where $\mathcal{F}^t = \sigma(R_1, \ldots, R_t, C_1, \ldots, C_t, K_1, \ldots, K_t)$.*

Detailed proofs of these lemma may be found in [18] and [30].

At all times $T$ we have

$$\begin{aligned}
\text{sFDR}_\delta(T) &= \mathbf{E}\left[\frac{V_\delta(T)}{R_\delta(T) + \eta}\right] \\
&= \mathbf{E}\left[\sum_{t=1}^T \frac{\delta^{T-t} R_t \mathbf{1}\{t \in \mathcal{H}^0\}}{R_\delta(T) + \eta}\right] \\
&\leq \mathbf{E}\left[\sum_{t=1}^T \frac{\delta^{T-t} R_t}{R_\delta(T) + \eta}\right].
\end{aligned} \tag{11}$$

So if we prove that the sum in the expectation is less than $\alpha$ at time $T$ then the $\text{sFDR}_\delta(T)$ is controlled. We show below that this is the case for LORD, SAFFRON and ADDIS. To do so, we let $\mathcal{R}(T) = \{t \in [T] \mid R_t = 1\}$ denote the set containing the times of rejections until step $T$.

### C.1 Proof of Proposition 1

The mapping $h(R_{1:T}) = \sum_{t=1}^T \delta^{T-t} R_t + \eta$ is coordinate-wise non-decreasing. If we use it to apply Lemma C.1 to the quantity in Inequality (11) we find that

$$\begin{aligned}
\text{sFDR}_\delta(T) &\leq \mathbf{E}\left[\sum_{t=1}^T \frac{\delta^{T-t} \alpha_t}{R_\delta(T) + \eta}\right] \\
&= \mathbf{E}\left[\widehat{\text{FDR}}_{\text{LORD}}^\delta(T)\right].
\end{aligned}$$

This shows that if the thresholds are chosen such that $\widehat{\text{FDR}}_{\text{LORD}}^\delta(T) \leq \alpha$ at time $T$ then $\text{sFDR}_\delta$ is controlled.

Let us now show that the special instance

$$\alpha_t = \alpha \eta \tilde{\gamma}_t + \alpha \sum_j \delta^{t-\rho_j} \gamma_{t-\rho_j}$$

satisfies this property. This is equivalent to show that the quantity

$$P(T) = \alpha(R_\delta(T) + \eta) - \sum_{t=1}^T \delta^{T-t} \alpha_t$$

is non-negative for all $T$. We find that

$$P(T) = \alpha\left(\sum_{t=1}^{T} \delta^{T-t} R_t + \eta\right) - \alpha \sum_{t=1}^{T}\left(\delta^{T-t}\eta\tilde{\gamma}_t + \sum_j \delta^{t-\rho_j}\gamma_{t-\rho_j}\right)$$

$$= \alpha\eta\left(1 - \sum_{t=1}^{T}\delta^{T-t}\tilde{\gamma}_t\right) + \alpha \sum_{t\in\mathcal{R}(T)} \delta^{T-t}\left(1 - \sum_{j=1}^{T-t}\gamma_j\right)$$

$$\geq 0,$$

where the inequality comes from the fact that $\{\gamma_t\}_{t=1}^{\infty}$ sums to 1 and that $\sum_{t=1}^{T}\delta^{T-t}\tilde{\gamma}_t \leq 1$.

## C.2 Proof of proposition 2

We let $\mathcal{G}^t = \sigma\left(R_1, \ldots, R_{t-L}\right)$ denote the non-conflicting filtration, *i.e.*, the information known at time $t$ which is not in conflict with the current $p$-value (it is indeed a filtration because $t - L$ is non-decreasing). In particular $\alpha_t$ is measurable with respect to $\mathcal{G}^{t-1}$. We write the oracle here again for convenience:

$$\widehat{\mathrm{FDP}}^{\delta}_{\mathrm{dep}}(T) = \frac{\sum_{t\leq T} \delta^{T-t}\alpha_t}{R_\delta(T) + \eta}.$$

The proof is essentially based on the analysis in [34]. We have

$$\mathbf{E}[V_\delta(T)] = \mathbf{E}\left[\sum_{t=1}^{T} \delta^{T-t} R_t \mathbf{1}\{t \in \mathcal{H}^0\}\right]$$

$$\leq \mathbf{E}\left[\sum_{t=1}^{T} \delta^{T-t} R_t\right]$$

$$= \sum_{t=1}^{T} \delta^{T-t}\mathbf{E}[R_t]$$

$$= \sum_{t=1}^{T} \delta^{T-t}\mathbf{E}\left[\mathbf{E}[R_t \mid \mathcal{G}^{t-1}]\right]$$

$$\leq \sum_{t=1}^{T} \delta^{T-t}\mathbf{E}[\alpha_t]$$

$$= \mathbf{E}\left[\sum_{t=1}^{T} \delta^{T-t}\alpha_t\right],$$

where we used the fact that $R_t$ is $\mathcal{G}^{t-1}$ measurable.

Now assume that $\widehat{\mathrm{FDP}}^{\delta}_{\mathrm{dep}}(T) \leq \alpha$. It follows that

$$\mathbf{E}[V_\delta(T)] \leq \alpha\mathbf{E}\left[R_\delta(T) + \eta\right]$$

and we conclude that $\mathrm{mFDR}_\delta(T) \leq \alpha$.

Let us now show that the special instance

$$\alpha_t = \alpha\eta\tilde{\gamma}_t + \alpha\sum_j \delta^{t-\rho_j}\gamma_{t-\rho_j-L}.$$

satisfies the above property. This is equivalent to show that the quantity

$$P(T) = \alpha(R_\delta(T) + \eta) - \sum_{t=1}^{T} \delta^{T-t}\alpha_t$$

is non-negative for all $T$. We compute that

$$P(T) = \alpha\left(\sum_{t=1}^{T} \delta^{T-t}R_t + \eta\right) - \alpha\sum_{t=1}^{T}\delta^{T-t}\left(\eta\tilde{\gamma}_t + \sum_j \delta^{t-\rho_j}\gamma_{t-\rho_j-L}\right)$$

$$= \alpha\eta\left(1 - \sum_{t=1}^{T}\delta^{T-t}\tilde{\gamma}_t\right) + \alpha\sum_{t=1}^{T}\delta^{T-t}\left(R_t - \sum_j \delta^{t-\rho_j}\gamma_{t-\rho_j-L}\right)$$

$$= \alpha\eta\left(1 - \sum_{t=1}^{T}\delta^{T-t}\tilde{\gamma}_t\right) + \alpha\sum_{t\in\mathcal{R}(T)}\delta^{T-t}\left(1 - \sum_{j=1}^{T-t-L}\gamma_j\right)$$

$$\geq 0.$$

## C.3  Proof of Proposition 3

For ADDIS the filtration is given by $\mathcal{F}^t = \sigma(R_{1:t}, C_{1:t}, K_{1:t})$. From Inequality (11) we derive

$$\text{sFDR}_\delta(T) \leq \mathbf{E}\left[\sum_{t=1}^{T}\frac{\delta^{T-t}R_t}{R_\delta(T)+\eta}\right]$$

$$= \sum_{t=1}^{T}\mathbf{E}\left[\mathbf{E}\left[\frac{\delta^{T-t}R_t}{R_\delta(T)+\eta} \mid K_t = 1, \mathcal{F}^{t-1}\right]\mathbf{Prob}\{K_t = 1 \mid \mathcal{F}^{t-1}\}\right] \qquad (12)$$

The mapping $h(R_{1:T}) = \sum_{t=1}^{T}\delta^{T-t}R_t + \eta$ is coordinate-wise non-decreasing. If we use it to apply Lemma C.2 to $\mathbf{E}\left[\frac{\alpha_t\mathbf{1}\{\lambda_t < p_t \leq \tau_t\}}{(\tau_t-\lambda_t)h(R_{1:T})} \mid \mathcal{F}^{t-1}, K_t = 1\right]$ Inequality (12) we find

$$\text{sFDR}_\delta(T) \leq \sum_{t=1}^{T}\mathbf{E}\left[\mathbf{E}\left[\delta^{T-t}\frac{\alpha_t\mathbf{1}\{\lambda_t < p_t \leq \tau_t\}}{(\tau_t-\lambda_t)(R_\delta(T)+\eta)} \mid \mathcal{F}^{t-1}, K_t = 1\right]\mathbf{Prob}\{K_t = 1 \mid \mathcal{F}^{t-1}\}\right]$$

$$= \sum_{t=1}^{T}\mathbf{E}\left[\delta^{T-t}\frac{\alpha_t\mathbf{1}\{\lambda_t < p_t \leq \tau_t\}}{(\tau_t-\lambda_t)(R_\delta(T)+\eta)}\right]$$

$$= \mathbf{E}\left[\widehat{\text{FDR}}^\delta_{\text{ADDIS}}(T)\right].$$

This shows that if the thresholds are chosen such that $\widehat{\text{FDR}}^\delta_{\text{ADDIS}}(T) \leq \alpha$ at time $T$ then the $\text{sFDR}_\delta(T)$ is controlled.

Let us now show that the special instance

$$\alpha_t = \alpha(\tau - \lambda)\left(\eta\tilde{\gamma}_{S_0(t)} + \sum_j \delta^{t-\rho_j}\gamma_{S_j(t)}\right) \wedge \lambda.$$

satisfies this property, where we define

$$S_j(t) = \mathbf{1}\{t > \rho_j\} + \sum_{i=\rho_j+1}^{t-1}\mathbf{1}\{\lambda < p_i \leq \tau\}$$

$$= \mathbf{1}\{t > \rho_j\} + \sum_{i=\rho_j+1}^{t-1}(1 - C_i)K_i.$$

This is equivalent to show that the quantity

$$P(T) = \alpha(R_\delta(T) + \eta) - \sum_{t=1}^{T}\frac{\delta^{T-t}\alpha_t\mathbf{1}\{\lambda < p_t \leq \tau\}}{(\tau - \lambda)}$$

is non-negative for all $T$. We find that

$$P(T) \geq \alpha \left( \sum_{t=1}^{T} \delta^{T-t} R_t + \eta \right) - \alpha \sum_{t=1}^{T} \delta^{T-t} \mathbf{1}\{\lambda < p_t \leq \tau\} \Big( \eta \tilde{\gamma}_{S_0(t)} + \sum_j \delta^{t-\rho_j} \gamma_{S_j(t)} \Big)$$

$$= \alpha \eta \left( 1 - \sum_{t=1}^{T} \delta^{T-t} (1 - C_t) K_t \tilde{\gamma}_{S_0(t)} \right) + \alpha \sum_{t=1}^{T} \delta^{T-t} \Big( R_t - (1 - C_t) K_t \sum_j \delta^{t-\rho_j} \gamma_{S_j(t)} \Big)$$

$$\geq \alpha \eta \left( 1 - \sum_{t=1}^{S_0(t)} \delta^{T-t} \tilde{\gamma}_t \right) + \alpha \sum_{t \in \mathcal{R}(T)}^{T} \delta^{T-t} \Big( 1 - \sum_{j=1}^{S_j(t)} \gamma_j \Big)$$

$$\geq 0.$$

Statements for SAFFRON are naturally obtained by setting $\tau_t = 1$ for all $t$.

# D   Adapting algorithms to standard FDR

We show here how to adapt the aforementioned algorithms to control the memory decay FDR without smoothing, that is,

$$\mathrm{FDR}_\delta(T) = \mathbf{E} \left[ \frac{V_\delta(T)}{R_\delta(T) \vee 1} \right].$$

To do so we define new oracles where the only difference is the denominator.

**LORD.**   For LORD we define the oracle

$$\widehat{\mathrm{FDP}}^{\delta}_{\mathrm{LORD}}(T) = \frac{\sum_{t \leq T} \delta^{T-t} \alpha_t}{R_\delta(T) \vee 1}.$$

Picking thresholds $\{\alpha_t\}_{t=1}^{\infty}$ such that $\widehat{\mathrm{FDP}}^{\delta}_{\mathrm{LORD}}(T) \leq \alpha$ ensure that $\mathrm{FDR}_\delta(T) \leq \alpha$. The proof for this statement is very similar as the one in Appendix C.1.

As a special instance, we pick $w_0 \in (0, \alpha)$ and define

$$\alpha_t = w_0 \tilde{\gamma}_t + (\alpha - w_0) \sum_j \delta^{t-\rho_j} \gamma_{t-\rho_j}. \tag{13}$$

We can check that $\widehat{\mathrm{FDP}}^{\delta}_{\mathrm{LORD}}(T) \leq \alpha$ at all time $T$ which shows that $\mathrm{FDR}_\delta$ is controlled. Notice that the thresholds are lower-bounded by $w_0(1 - \delta)$.

**SAFFRON and ADDIS.**   For ADDIS we define the oracle

$$\widehat{\mathrm{FDP}}^{\delta}_{\mathrm{ADDIS}}(T) = \frac{\sum_{t \leq T} \delta^{T-t} \alpha_t \frac{\mathbf{1}\{\lambda_t < p_t \leq \tau_t\}}{\tau_t - \lambda_t}}{R_\delta(T) \vee 1}.$$

Here again, if we pick the threshold such that $\widehat{\mathrm{FDP}}^{\delta}_{\mathrm{ADDIS}}(T) \leq \alpha$ then $\mathrm{FDR}_\delta(T) \leq \alpha$ and the proof is very similar to the one in Appendix C.3.

As a special instance, we pick $\tau > \lambda$, $w_0 \in (0, \alpha)$ and define

$$\alpha_t = (\tau - \lambda) \Big( w_0 \tilde{\gamma}_{S_0(t)} + (\alpha - w_0) \sum_j \delta^{t-\rho_j} \gamma_{S_j(t)} \Big) \wedge \lambda.$$

We can check that at any time $T$,

$$\widehat{\mathrm{FDP}}^{\delta}_{\mathrm{ADDIS}}(T) \leq \alpha$$

which shows that $\mathrm{FDR}_\delta$ is controlled. Again, SAFFRON is just a special case where $\tau_t = 1$ for all $t$. Thresholds are lower-bounded by $(\tau - \lambda) w_0 (1 - \delta) \wedge \lambda$.

**Local dependence.**   For the local dependency setup we can define the thresholds

$$\alpha_t = w_0 \tilde{\gamma}_t + (\alpha - w_0) \sum_j \delta^{t-\rho_j - L} \gamma_{t-\rho_j - L}.$$