# OpenReview forum: "Online false discovery rate control for anomaly detection in time series"
_NeurIPS.cc/2021/Conference — NeurIPS 2021 Poster_

### Official Review · Reviewer_4PWU · 2021-07-16

**Rating:** 6
**Confidence:** 3

**Summary:**

This paper address online false discovery rate (FDR $\simeq 1-\mathrm{Precision}$) control in anomaly detection, where we obtain observations sequentially and immediately decide whether we classify them into anomaly or not.
The authors adopted smoothed FDR (sFDR) for criteria and then derived rejection thresholds that offer high detection power (Recall) for anomalies without shrinking its detection threshold to zero while maintaining target sFDR. It is made possible by the fact that FDR for no rejection is defined by the explicit parameter $\eta$ in the sFDR.
They also proposed a way to handle local dependency in time-series with a similar procedure.
Experimental results on synthetic datasets demonstrate that the proposed method performs better than existing methods, especially when an anomaly is very rare.

**Limitations And Societal Impact:**

They addressed the limitations and potential negative social impact of their work.

**Main Review:**

### Originality:
* This work proposed novel rejection thresholds, $\alpha_t$s, in Eqs. (8) and (9) for anomaly detection with a capability of online FDR control. It is derived from well-studied sFDR [18].

### Quality:
* Related work is well-written and would be informative for many readers.

* Experimental evaluation is weak.
    - It is only on synthetic datasets.
    - They even did not conduct experiments on time-series or sequential data.
    - It seems that in cases where anomalies are rare, the system worked well. However, in cases where anomalies are more common, the performance deteriorates. I would still like to see how the proposed method works in real data, especially in more realistic scenario where we consider decay and local dependency.
    - The authors stated that "The appendix contains further experiments taking local dependency into account." in l.288, but there would be no such experiment in the appendix. I guess that they had tried to add experiments after submission but they could not. It could be unfair.

### Clarity:
* They did not clearly state the contributions of this paper. It would be better to state them.

* It would be better to cite conference/journal not arXiv, such as [23] is published in Neurips2019 and [32] is published in AISTATS2020.

* In l.142, what is "reward" in this context?

* In l.267, sFDR would be mFDR?

* Figure 2 is not mentioned in the main text. Also, in general, it is better to locate figures on top.

### Significance:
* The proposed method could be practically imoportant since it offers high recall for anomalies while maintaining target precision. However, the significance on real data is not clear because they did not provide experiments on real datasets.

**Time Spent Reviewing:**

6

---

> ### Author Response · Authors · 2021-08-10
> **Clarity**
>
> We will add our contributions more clearly in the introduction. More precisely, we will emphasize that 1) we propose memory-decay variants to existing FDR rules and show that they prevent alpha-death, 2) include memory-decay to local dependency rules in order to handle dependent p-values, 3) we apply FDRC to anomaly detection in time series.
>
> We will also work on the citations to include the proper venues instead of the arxiv versions.
>
> Regarding your other comments:
> - "In l.142, what is "reward" in this context?"
> The word “reward” in line 142 refers to the quantity phi_t defined in line 141: the amount of wealth that is earned back when a rejection occurs. A clearer formulation could be “FDR is controlled only if the quantities psi_t are bounded adequately”. We will clarify this in the manuscript.
> - "In l.267, sFDR would be mFDR?"
> Thank you for catching this. Indeed, it should be the modified FDR with the smoothing parameter defined after line 263.
> - "Figure 2 is not mentioned in the main text. Also, in general, it is better to locate figures on top."
> Thank you for pointing this out and we will refer to the figure from the main text.

---

> > ### Comment · Reviewer_4PWU · 2021-08-29
> > **Thak you for your response**
> >
> > I would like to see the difference and advancement over the method in the paper [18], which used sFDR already. You mentioned that the proposed method is inspired by the paper [18], so it is better to clarify the difference.

---

> ### Author Response · Authors · 2021-08-10
> **Rare/common anomalies**
>
> The decay parameter (delta) should depend on the frequency of anomalies. We tuned mostly for rare anomalies regimes because this is the main obstruction to existing FDRC techniques. When anomalies are more common, past decisions should shrink much slower and delta should be closer to 1. In this case, the proposed methods should perform as well as other benchmarked algorithms.

---

> ### Author Response · Authors · 2021-08-10
> **Additional experiments**
>
> We thank the reviewer for his comments.
>
> __Experimental results & Significance__
>
> We have conducted a real-world experiment given yours and the other reviewers comments using the server machine data set [1]. We describe it further below. The results of this experiment are available in [this double-blind preserving repository]( https://anonymous.4open.science/r/fdrc_rebuttal-experiments/readme.txt). Note that anomalies are not rare in the SMD dataset, 4.16% are labelled as anomalies, which limits the benefits of memory decay relative to baseline methods. We also acknowledge and share the concern of reviewer SufA regarding the relevance of the SMD data set, despite it being standard in the literature.
>
> __Missing experiment for local dependency__
>
> We apologize for this oversight in the manuscript: indeed, the experiment was not part of the appendix. We note that as part of the real-world experiments, we already have some locally dependent data (because the data consists of time series with autocorrelation) and we will update this review-thread with dedicated experiments on synthetic data. We obtained precision-recall curves that show that the proposed methods perform favorably compared to the non-decay versions.
>
> We also repeated the synthetic data experiment from our submitted paper. But this time we generated locally dependent anomalies using a simple Markov chain with two states. We observed that the memory-decay algorithms maintain a high power and still control the memory-decay FDR.
>
> [1] Su Y., Zhao Y., Niu C., Liu R. Sun W., Pei D., "Robust Anomaly Detection for Multivariate Time Series through Stochastic Recurrent Neural Netw ork" ,  ​​KDD '19: Proceedings of the 25th ACM SIGKDD International Conference on Knowledge Discovery & Data MiningJuly 2019 Pages 2828–2837https://doi.org/10.1145/3292500.3330672

---

### Official Review · Reviewer_xktj · 2021-07-16

**Rating:** 7
**Confidence:** 3

**Summary:**

This research seeks to control the false-discovery rate, while simultaneously avoiding alpha-death, where the alpha values are shrunk to near zero for multiple-hypothesis testing with extremely rare events. The authors propose a method based on memory-decay that is extensible to other algorithms of the alpha-investing class.

**Limitations And Societal Impact:**

The authors did an adequate job of discussing the limitations of their research.There were no obvious potential negative societal impacts that I believe the authors missed.

**Main Review:**

The paper was well structured in setting up the problem, detailing the previous work and limitations, providing a solution and an extension, and providing experimental results. The exeprimental results could be fleshed out more, although I have trouble deciding from where the extra room would have come, besides perhaps the "False discovery rate control" section or removing some of the formulations in the "Oracle rules" section.

Overall, I believe their extensible decay contributions to be useful to the field. I believe this research provides ample starting point for others, or the authors themselves, to explore.

**Time Spent Reviewing:**

5

---

> ### Author Response · Authors · 2021-08-10
> **Additional experiment on real-world data**
>
> We thank the reviewer for his positive review. We will add an experiment on a real-world data set (server machine data set [1]) given yours and the other reviewer’s comments. The results of this experiment are available in [this double-blind preserving repository]( https://anonymous.4open.science/r/fdrc_rebuttal-experiments/readme.txt). Note that anomalies are not rare in the SMD dataset, 4.16% are labelled as anomalies, which limits the benefits of memory decay relative to baseline methods. We also acknowledge and share the concern of reviewer SufA regarding the relevance of the SMD data set, despite it being standard in the literature.
>
> [1] Su Y., Zhao Y., Niu C., Liu R. Sun W., Pei D., "Robust Anomaly Detection for Multivariate Time Series through Stochastic Recurrent Neural Netw ork" ,  ​​KDD '19: Proceedings of the 25th ACM SIGKDD International Conference on Knowledge Discovery & Data MiningJuly 2019 Pages 2828–2837https://doi.org/10.1145/3292500.3330672

---

### Official Review · Reviewer_trQ6 · 2021-07-16

**Rating:** 6
**Confidence:** 2

**Summary:**

This paper discusses techniques by which some limitations of current online false discovery rate control algorithms can be overcome. Importantly, the sequential decision thresholds from current algorithms decay to zero in the absence of recent rejections of the null hypothesis -- a situation called alpha-death. A technique to overcome this is proposed.

================
Update: The scope of contribution has been clarified in the rebuttal. The authors address the lack of real-world experiments somewhat, but more is desired to improve the strength of the paper.

**Limitations And Societal Impact:**

Limitations have been discussed.

**Main Review:**

1. It was a bit hard to figure out the main novel contribution in this paper. It seems that the equation after line 216 (smoothed FDR) is the claimed contribution, but since the equation is not numbered, it does not come out clearly. Moreover, most equations on page 8 are also unnumbered and perhaps part of the contribution of this paper. All these are minor variations over existing techniques.

2. The experiments section (line 275) states that the paper does not aim to show that the technique results in yielding higher anomaly detection accuracy than alternatives. This is a serious drawback and coupled with the fact that no real-world datasets have been used perhaps the paper in its current state is only fit for a workshop rather than a full conference.

**Time Spent Reviewing:**

2

---

> ### Author Response · Authors · 2021-08-10
> **Clarity of main contribution**
>
> We will bring out this discussion better in the manuscript by adding a concise list of points in the introduction and proper numberings for the main equations (thank you for the pointer to this oversight).
>
> The smoothed FDR (equation after line 216) is not novel and was already studied (see for example [16]). We merely use it as a tool, and we also show that the results hold for the standard FDR in the appendix.
>
> Our first contribution is the “memory-decay” variants of existing online FDR control techniques (LORD, SAFFRON and ADDIS) in Section 4. These variants prevent alpha-death, and we provide theoretical guarantees that they control a memory-decay version of the FDR (proposition 1 for LORD and proposition 3 for SAFFRON and ADDIS in Appendix A). We propose some special instances of the algorithms and corresponding thresholds (equations 8 and 9). We think that alpha-death is a serious issue for existing techniques, especially in practical applications and in the context of anomaly detection. Amending the existing techniques fills an important gap in the literature. We think that extending existing techniques is beneficial as they were proven to be robust for FDR control. The principles may also be generalized to future techniques that could leverage memory decay.
>
> The second contribution is the adaptation of the local dependency framework to memory decay. This allows to use our methods for practical applications where p-values are in general not independent.
>
> The third contribution is the application of online false discovery rate control for anomaly detection tasks in time series. We show that we can overcome the challenges occurring when alternative hypotheses are exceedingly rare. We performed another experiment on real data (see below).

---

> ### Author Response · Authors · 2021-08-10
> **Experimental results**
>
> We recognize that we have not sufficiently highlighted the main benefit of our proposed method for anomaly detection tasks: allowing the user to target a specific FDR/precision ex-ante, before having observed any data or labels. Standard benchmarks focus on determining the best F1 score ex-post, after observing labels. This does not reflect the experience of a user who needs to decide on what threshold to apply for an anomaly detection task before running the experiment. Our method is unique in offering a way to target a specific precision from the onset. This is highly relevant for practical anomaly detection applications.
>
> In view of your comments and those of the reviewers, we focus on demonstrating that the novel capability that we provide users of anomaly detection methods does not come at the cost of a loss of accuracy relative to traditional, ex-post, measures. We have therefore done the following:
> 1) We have conducted an experiment on the server machine data set [1] for anomaly detection as a real-world experiment given your and other reviewers’ valid concerns and we will add this to the manuscript. In brief, alpha-death indeed occurs also on real-world data sets and our methods have a clear advantage over the non-decayed version. The results of this experiment are available in [this double-blind preserving repository]( https://anonymous.4open.science/r/fdrc_rebuttal-experiments/readme.txt). Note that anomalies are not rare in the SMD dataset, 4.16% are labelled as anomalies, which limits the benefits of memory decay relative to baseline methods. We also acknowledge and share the concern of reviewer SufA regarding the relevance of the SMD data set, despite it being standard in the literature.
> 2) We added a comparison against a fixed thresholding scheme that would be typically employed in practice for this data set and show that our decay results compare favorably.
>
> [1] Su Y., Zhao Y., Niu C., Liu R. Sun W., Pei D., "Robust Anomaly Detection for Multivariate Time Series through Stochastic Recurrent Neural Netw ork" ,  ​​KDD '19: Proceedings of the 25th ACM SIGKDD International Conference on Knowledge Discovery & Data MiningJuly 2019 Pages 2828–2837https://doi.org/10.1145/3292500.3330672

---

> > ### Comment · Reviewer_trQ6 · 2021-08-26
> > **Draft needs to be updated**
> >
> > For me to revise my scores, I would like to see the submission updated.

---

> > > ### Author Response · Authors · 2021-08-26
> > > **Unable to update the submission**
> > >
> > > It is unfortunately not possible to update the submission as far as we understand. We have shared supplementary results in the double-blind preserving repository linked above. They will be part of the updated paper. We will also incorporate our response to your comments regarding the clarity of the contribution (see "Clarity of main contribution" below) into the updated version.
> > >
> > > We will evaluate whether sharing an updated draft through a double-blind preserving channel is possible and does not constitute a breach of NeurIPS submission rules.

---

> > > > ### Comment · Reviewer_trQ6 · 2021-08-26
> > > > **No need to deviate from the NeurIPS submission rules.**
> > > >
> > > > I am updating my scores on the basis that the paper looks good other than the experiments section.

---

> ### Author Response · Authors · 2021-08-10
> **General comment**
>
> We thank the reviewer for his comments. We address them in the responses below.

---

### Official Review · Reviewer_SufA · 2021-07-19

**Rating:** 9
**Confidence:** 2

**Summary:**

This paper proposes novel rules for false discovery rate control (FDRC) for online anomaly detection in time series

**Main Review:**

This paper is slightly out of my area, so this will be a low confidence review. In particular, I cannot strongly comment on the novelty.

Having said that, this is clearly the best paper in my bunch (of seven). Well motivated, very well written and possessing deep insights in statistics.  The paper clearly identifies its limitations

The paper ends with “We hope to be able to follow up on this work once higher-quality public datasets become available.” I would love to see this (long journal) paper, I think it would be a very useful resource for the community. However, within the page limits available this current work makes a strong contribution.


**Time Spent Reviewing:**

2

---

> ### Author Response · Authors · 2021-08-10
> **Additional experiment on real-world data**
>
> We thank the reviewer for their positive comments and encouraging feedback. We have conducted an experiment with the server machine dataset [1] which we will add to the paper. The results of this experiment are available in [this double-blind preserving repository]( https://anonymous.4open.science/r/fdrc_rebuttal-experiments/readme.txt). Note that anomalies are not rare in the SMD dataset, 4.16% are labelled as anomalies, which limits the benefits of memory decay relative to baseline methods.
>
>
> [1] Su Y., Zhao Y., Niu C., Liu R. Sun W., Pei D., "Robust Anomaly Detection for Multivariate Time Series through Stochastic Recurrent Neural Netw ork" ,  ​​KDD '19: Proceedings of the 25th ACM SIGKDD International Conference on Knowledge Discovery & Data MiningJuly 2019 Pages 2828–2837https://doi.org/10.1145/3292500.3330672

---

> > ### Comment · Reviewer_SufA · 2021-08-10
> > **Thanks for being so responsive**
> >
> > Thanks for being so responsive
> >
> > However, you say.. "We have conducted an experiment with the server machine dataset [1]"
> > Noooooo!
> >
> > That datasets is a bit of a joke (as are most papers from that lab).
> >
> > Lets see this.
> > Get OmniAnomaly/ServerMachineDataset/test/machine-2-5.txt
> > Consider only dim 14, call it 'T'
> >
> > Now try this single line of matlab code
> >
> > >> my_labels =     T > mean(T) + (2* std(T));
> >
> > Now compare the labels we just made to the ground truth.
> >
> > ---
> > Or try this one, download machine-13-11.txt
> > Consider only dim 19, call it 'M19'
> > Now try this
> > >> my_labels = M19 < 0.01
> >
> > Now compare the labels we just made to the ground truth.
> >
> > ---
> > If I can solve all these problems with one line of code, maybe they are too easy

---

> > > ### Author Response · Authors · 2021-08-11
> > > **Point well taken!**
> > >
> > > We agree that this dataset and many other standard datasets are solvable with one-liners and therefore of moderate utility, despite being standard in benchmarking. Wu and Keogh (2020) is an enlightening read on this topic. We will add experiments with more informative datasets to the next revision of the paper, but such datasets are hard to source.

---

> > > > ### Comment · Reviewer_SufA · 2021-08-26
> > > > **I have some sympathy for "but such datasets are hard to source"**
> > > >
> > > > I have some sympathy for "but such datasets are hard to source", however, should we really have 100 papers on anomaly detection per year, and all tested on the same 3 or 4 worthless datasets?   How would we know if we are making progress?
> > > >
> > > > At the end of the day, if you cannot find datasets, you can make them.
> > > >
> > > > Wear an iphone in your back pocket, walk for five miles, at the 3 mile mark, have someone throw a heavy ball at you...
> > > >
> > > > Attach an iphone to the pedal of a bike, do ten laps of velodrome, on the fifth lap, do a wheelie
> > > >
> > > > If you are too busy for the above, get normal ECGs from [a]. Take an hours of joes heartbeats, and carefully edit in one of sues heartbeats, now you have some anomaly....
> > > >
> > > > My point is, you can work hard, and you can be creative.
> > > >
> > > > I dont think you appreciate how worthless these datasets are, if they are mislabeled, what does accuracy even mean?
> > > >
> > > >
> > > >
> > > > [a] https://archive.physionet.org/cgi-bin/atm/ATM

---

> > > > > ### Author Response · Authors · 2021-08-27
> > > > > **We agree with the recommendation and appreciate the suggestions.**
> > > > >
> > > > > Thank you for you comment, with which we agree.
> > > > >
> > > > > We are in the process of sourcing and labeling a large enough dataset from the AIOps space in order to have a more meaningful benchmark on real-world data. Thank you for pointing out the ECG data, we will explore this possibility as well. As you yourself point out, this is hard work. Creating, validating, and open-sourcing such a dataset is not something that we are able to do within the time imparted for this response period unfortunately. We are aiming at using it in an updated version of the paper though.
> > > > >
> > > > > I'd also like to point out that, in contrast to the vast majority of anomaly detection papers, we are not proposing a new anomaly detection model. We are instead proposing a model-agnostic method to make decision in online anomaly detection tasks based on a detector's output without observing labels. To the best of our knowledge, this is not a question frequently explored in the literature despite being of great practical importance.

---

### Author Response · Authors · 2021-08-26
**Thank you to the reviewers and chairs**

We would like to thank the reviewers again for the time they dedicated to reviewing our work and for their insightful comments. As the discussion period is almost over, we hope that we have adequately answered all the reviewers' comments. If further clarifications are needed, please feel do to let us know.

---

### Decision · Program_Chairs · 2021-09-27

**Decision:**

Accept (Poster)

**Comment:**

After the rebuttal phase, the paper now has only positive reviews, and as such could
be accepted. The main issues found during the review phase are the somewhat unclear novelty
and the insufficient experiments. After carefully reading the paper again, I need to add
another concern: the independence assumption of the p-values. While the authors try to
address this issue in Section 5, their approach is somewhat limited:

- It is assumed that sufficiently far away p-values are independent, this is measured by the
  distance L in time, see (10). However, it remains unclear whether and when this assumption is satisfied,
  and the authors do not discuss this issue.

- The only proposed method working for (10) needs to know L, which again sounds rather unrealistic.

In summary, despite having a good average rating and a positive rebuttal phase, I personally see
this as a true borderline case.